# The vacuole/lysosome is required for cell-cycle progression

Yui Jin, Lois S Weisman*

Life Sciences Institute, Department of Cell and Developmental Biology, University of Michigan, Ann Arbor, United States

**Abstract** Organelles are distributed to daughter cells, via inheritance pathways. However, it is unclear whether there are mechanisms beyond inheritance, which ensure that organelles are present in all cells. Here we present the unexpected finding that the yeast vacuole plays a positive essential role in initiation of the cell-cycle. When inheritance fails, a new vacuole is generated. We show that this occurs prior to the next cell-cycle, and gain insight into this alternative pathway. Moreover, we find that a combination of a defect in inheritance with an acute block in the vacuole biogenesis results in the loss of a functional vacuole and a specific arrest of cells in early G1 phase. Furthermore, this role for the vacuole in cell-cycle progression requires an intact TORC1-*SCH9* pathway that can only signal from a mature vacuole. These mechanisms may serve as a checkpoint for the presence of the vacuole/lysosome.

## Introduction

Organelles are essential for cellular functions, and organelle inheritance is likely a major pathway that ensures the presence of organelles in all cells. A requirement for the inheritance of the mammalian Golgi is well established due in part to the elaborate changes in architecture that occur during interphase vs mitosis (*Shorter and Warren, 2002*). For other organelles, which are constitutively dispersed, organized mechanisms for their inheritance remain less clear. The budding yeast *Saccharomyces cerevisiae* provides an excellent model to study the spatial and temporal control of organelle inheritance, in part because its cell division is asymmetric. This asymmetric division requires active organelle transport in each cell-cycle. In budding yeast, most of the organelles are transmitted from mother to daughter cells (*Fagarasanu and Rachubinski, 2007*). These include the vacuole/lysosome, mitochondria, the endoplasmic reticulum, peroxisomes, secretory vesicles and late-Golgi elements. Transport of these organelles starts in G1 phase and occurs in coordination with the cell-cycle. However, it is unclear whether there are mechanisms that guarantee the presence of organelles prior to the next round of cell division. Here we present the unexpected finding that the presence of the vacuole is ensured because the vacuole plays an essential role in the initiation of the cell-cycle.

During cell division in budding yeast, the daughter cell inherits a vacuole from the mother cell (*Weisman et al., 1987*). The vacuole is transported by a vacuole transport complex, composed of the myosin V motor Myo2, the vacuole membrane anchored protein Vac8, and an adaptor protein Vac17 that links Myo2 and Vac8 (*Catlett and Weisman, 1998*; *Wang et al., 1998*; *Ishikawa et al., 2003*; *Tang et al., 2003*). Vacuole inheritance is initiated in G1 phase via Cdk1/Cdc28, which regulates the formation of the vacuole transport complex (*Peng and Weisman, 2008*). After formation of the complex, Myo2 moves the vacuole to the daughter cell along actin cables (*Hill et al., 1996*). At the end of the cell-cycle, vacuole transport is terminated by ubiquitylation of Vac17, which is then degraded by the 26S proteasome (*Yau et al., 2014*). Notably, Myo2 also delivers other cargoes including mitochondria, peroxisomes, secretory vesicles, late-Golgi elements, and astral microtubules. Myo2 binds to each cargo via cargo specific adaptors, which attach to the globular tail domain of

*For correspondence:
lweisman@umich.edu

Competing interests: The authors declare that no competing interests exist.

**eLife digest** Animals, fungi and other eukaryotes have cells that are divided into sub-compartments that are called organelles. Each type of organelle serves a specific purpose that is essential for the life of the cell. Yeast cells have a large organelle called a vacuole; the inside of the vacuole is acidic and contains enzymes that can break down other molecules.

Previous studies have shown that when a budding yeast cell buds to produce a new daughter cell, a process ensures that some of the mother's vacuole is transferred to its daughter. However, yeast mutants that fail to inherit some of their mother's vacuole can still survive. This is because an 'alternative' mechanism allows the newly forming daughter to generate its own vacuole from scratch.

Jin and Weisman now unexpectedly show that a new daughter cell cannot become a mother cell until its new vacuole is formed. The experiments made use of yeast mutants that were defective in the 'inheritance' mechanism, and double mutants that were defective in both the inheritance and alternative mechanisms. The experiments also revealed that a signal from the vacuole is required before the yeast cell's nucleus can start the cycle of events that lead to the cell dividing. Jin and Weisman suggest that this newly identified communication between the vacuole and the nucleus may help to ensure that critical organelles are present in all cells.

Though it remains unclear why the yeast vacuole is critical for a cell to divide, these findings suggest that the mammalian lysosome (which is similar to the yeast vacuole) may perform a similar critical role in mammals. If this is the case, then understanding how these organelles communicate with the nucleus may provide new insights into how to prevent the uncontrolled growth of tumors and cancer.

Myo2 (*Yin et al., 2000*; *Itoh et al., 2002*; *Boldogh et al., 2004*; *Itoh et al., 2004*; *Fagarasanu et al., 2006*; *Arai et al., 2008*; *Lipatova et al., 2008*; *Jin et al., 2011*; *Santiago-Tirado et al., 2011*; *Eves et al., 2012*; *Chernyakov et al., 2013*). Moreover, some of the regulatory pathways for vacuole transport are also utilized by other Myo2 cargoes (*Moore and Miller, 2007*; *Peng and Weisman, 2008*; *Fagarasanu et al., 2009*; *Jin et al., 2009*; *Yau et al., 2014*).

Many of the proteins involved in vacuole inheritance are conserved among several species, which suggests that vacuole inheritance confers a selective advantage (*Mast et al., 2012*). These observations suggest that the vacuole plays essential roles. Surprisingly, mutations that block vacuole inheritance do not have a notable impact on cell viability (*Catlett and Weisman, 1998*; *Ishikawa et al., 2003*). Indeed, previous studies suggest that new vacuole synthesis occurs in the absence of vacuole inheritance (*Weisman et al., 1990*; *Gomes De Mesquita et al., 1997*), however at the time of those studies, there were no suitable methods to distinguish an old vacuole from newly formed vacuoles. Moreover the origin of the new vacuole was unknown. Importantly it was not clear how many pathways would need to be blocked in order to prevent vacuole biogenesis. Note that vacuole biogenesis utilizes at least three direct transport pathways: autophagy/Cvt (from the cytoplasm), AP-3/ALP (from the Golgi), and CPY (from the MVB/endosome) pathways (*Bryant and Stevens, 1998*; *Hecht et al., 2014*).

## Results and discussion

To test when and where a new vacuole is generated in the absence of vacuole inheritance, we monitored for the presence of a vacuole using two markers, Vph1 and FM4-64. Vacuoles were detected using GFP fused to the integral vacuole membrane protein Vph1, a $V_0$ subunit of the vacuolar ATPase (*Manolson et al., 1992*). The presence of inherited vacuoles or old vacuoles were specifically assessed via pulse chase experiments with the vital fluorophore FM4-64 (*Vida and Emr, 1995*). Exogenously added FM4-64 binds to the plasma membrane, is internalized by endocytosis and delivered to the vacuole. After a chase of one doubling time, all of the FM4-64 is trapped on the vacuole membrane. In wild-type cells, the vacuole is inherited and FM4-64 is distributed between the mother and daughter vacuole (*Wang et al., 1996*), and Vph1-GFP and FM4-64 always colocalize (*Figure 1A*, top panels). In contrast, in the vacuole inheritance mutant *vac17Δ*, FM4-64 is retained in the mother cell (*Ishikawa et al., 2003*). Interestingly in *vac17Δ*, buds contained small Vph1-GFP positive vacuoles that lack FM4-64 (*Figure 1A*; open white allow heads, see also [*Anand et al., 2009*]).

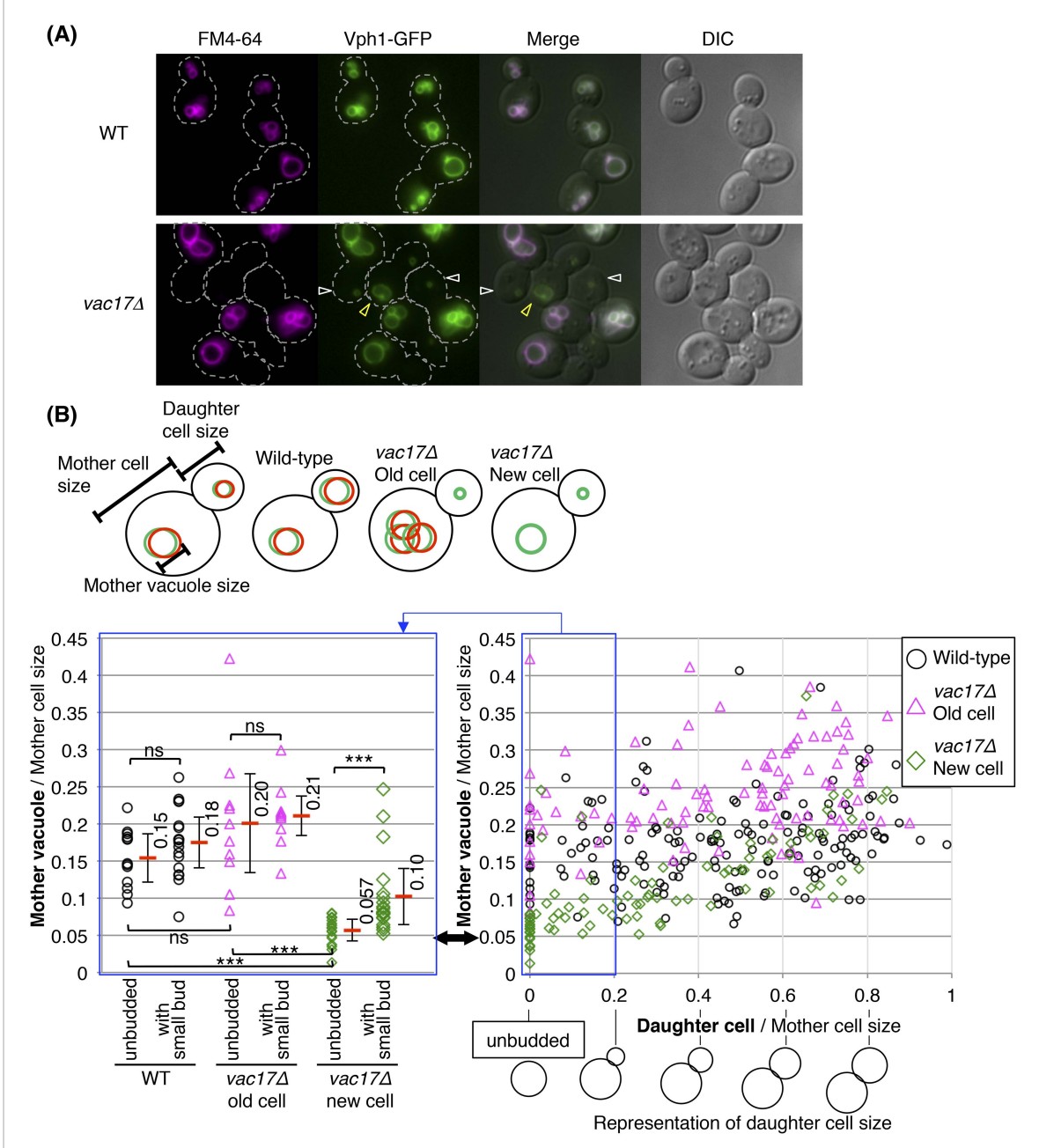

**Figure 1**. Yeast require vacuoles of a specific size prior to formation of a bud. (**A**) Wild-type and *vac17Δ* cells which express Vph1-GFP from its endogenous locus, were pulse labeled with the vacuole specific dye FM4-64. Wild-type cells have both FM4-64 and Vph1-GFP signals in both mother and daughter cells. *vac17Δ* cells have both Vph1-GFP and FM4-64 on the vacuole in old mother cells, however the daughter cells solely have a Vph1-GFP labeled vacuole. White arrowheads; new vacuoles in daughter cells. Yellow arrowheads; new vacuoles in new mother cells. Dashed line; outline of cells. (**B**) (Left panel) A new cell does not form a bud until its vacuole reaches a specific size. Graph indicates the vacuole diameter of wild-type, *vac17Δ* old cell and *vac17Δ* new cells with no bud and mother cells with a small bud (less than 20% of diameter of the mother cell). Cell/vacuole diameter was measured by ImageJ. Each cell/vacuole diameter was normalized to its mother cell diameter. Black arrow; minimum size of mother vacuoles in cells with a bud. Average in each category (red bar). Error bar; standard deviation (SD). Not a significant difference; ns, p-value > 0.10. A statistically significant difference; *** (p-value < 1 × 10⁻³). (Right panel) The vacuoles of the new mother cells of *vac17Δ* grow faster than vacuoles in either a wild-type or *vac17Δ* old mother cell. Scatter plots of bud sizes and mother vacuole sizes.

The following figure supplement is available for figure 1:

**Figure supplement 1**. Yeast require vacuoles of a specific size prior to formation of a bud.

This observation indicates that there is new vacuole synthesis in the absence of vacuole inheritance, and that the new synthesis initiates in the bud.

## Yeast generates vacuoles of a specific size prior to formation of a bud

The new mother cells of *vac17Δ*, which lack FM4-64 (*Figure 1A*; open yellow arrowheads), had larger vacuoles compared to the newly formed vacuoles in the large buds. These findings suggest that the newly generated vacuoles continue to grow. To establish the relationship between the size of the newly formed vacuoles with the size of unbudded cells or mother cells, we measured vacuole size vs cell size in wild-type and *vac17Δ* cells. In wild-type cells, the vacuole diameter showed a linear relationship with cell diameter (*Figure 1—figure supplement 1A,B*). This is consistent with a previous study that demonstrated that vacuole volume correlates with cell volume (*Chan and Marshall, 2014*). Notably, we found that in the *vac17Δ* mutant, the vacuoles grew prior to the generation of a bud (*Figure 1B*, black arrow). In unbudded new daughters of the *vac17Δ* mutant, the vacuole diameter relative to cell diameter was smaller than that of wild-type and *vac17Δ* unbudded old mother cells (*Figure 1B*, left panel). Notably, the average diameter of the vacuole was only 5.7($\pm$1.5)% of the cell diameter in new unbudded *vac17Δ* cells. However, after production of a small bud, the average mother cell vacuole diameter was 10($\pm$3.8)% of the cell diameter. In contrast, there was no significant increase in the relative percent diameter of the vacuoles in wild-type and *vac17Δ* old mother cells with or without a small bud. These observations show that the vacuoles in the new daughter cells grow to a minimum size prior to producing a bud. This growth occurs either because a minimum vacuole size is required and/or because the vacuole needs to mature prior to the generation of a bud.

Note that the vacuoles in *vac17Δ* new mothers continued to grow, and grew faster than vacuoles in either wild-type or *vac17Δ* old mother cells (*Figure 1—figure supplement 1C*, green line vs the black/pink lines). These observations suggest that cells actively synthesize new vacuoles in the absence of inherited vacuoles, and further suggest that there are mechanisms that regulate vacuole size in proportion to cell size.

In addition, the average vacuole diameter relative to cell diameter of all the *vac17Δ* old mother cells was larger than that of wild-type and *vac17Δ* new mother cells (*Figure 1—figure supplement 1D*). This suggests that vacuole inheritance is also important for regulating vacuole size in the mother cell.

## The vacuole is required for cell growth

We observed that all *vac17Δ* mother cells and unbudded cells have a vacuole as defined by the presence of Vph1-GFP. This strongly suggests that the vacuole is required for cell growth and viability. If this were true, then a combination of a vacuole inheritance defect with an additional defect in the synthesis of a new vacuole would render the cell inviable. Similarly, an additional defect in vacuole function(s) that are required for bud emergence would result in non-viable cells (*Figure 2A*). Indeed, a high-throughput screen suggested that over twenty genes might be synthetically lethal with the *vac17Δ* mutant (*Costanzo et al., 2010*). We individually tested double mutants of *vac17Δ* with each of the previously proposed candidates that are not essential genes, and found that the double mutants, *vac17Δ pep12Δ* and *vac17Δ vps45Δ* displayed synthetic growth defects (*Figure 2B,C*). Importantly *pep12Δ* and *vps45Δ* were also synthetically lethal with additional mutants defective in vacuole inheritance, *vac8Δ* and *myo2-N1304D* mutants (*Figure 2—figure supplement 1A–D*). The corresponding wild-type genes, *PEP12* and *VPS45*, likely play a critical role in the generation of a new vacuole.

*PEP12* encodes a t-SNARE, and *VPS45* encodes a Sec1/Munc18 protein. These proteins function together in the vacuole-protein-sorting pathway from endosomes to the vacuole (*Becherer et al., 1996*; *Burd et al., 1997*). To test whether the growth defects of the *vac17Δ pep12Δ* double mutant are due to a defect in forming new vacuoles, we monitored vacuoles using two markers, Vph1-GFP and CMAC, a small molecule that is taken into the lumen of the vacuole (*Stefan and Blumer, 1999*). Wild-type and *vac17Δ* cells showed normal localization of Vph1-GFP and CMAC (*Figure 2D,E*). However, single *pep12Δ* cells, which have a defect in protein sorting to the vacuole, and a partial defect in vacuole inheritance (*Raymond et al., 1992*), showed an abnormal distribution of Vph1 (*Piper et al., 1997*), but not CMAC (*Figure 2D,E*). The *vac17Δ pep12Δ* double mutant cells showed defects in the localization of Vph1 and CMAC (*Figure 2D,E*). These findings suggest that the double mutant does not generate normal vacuoles.

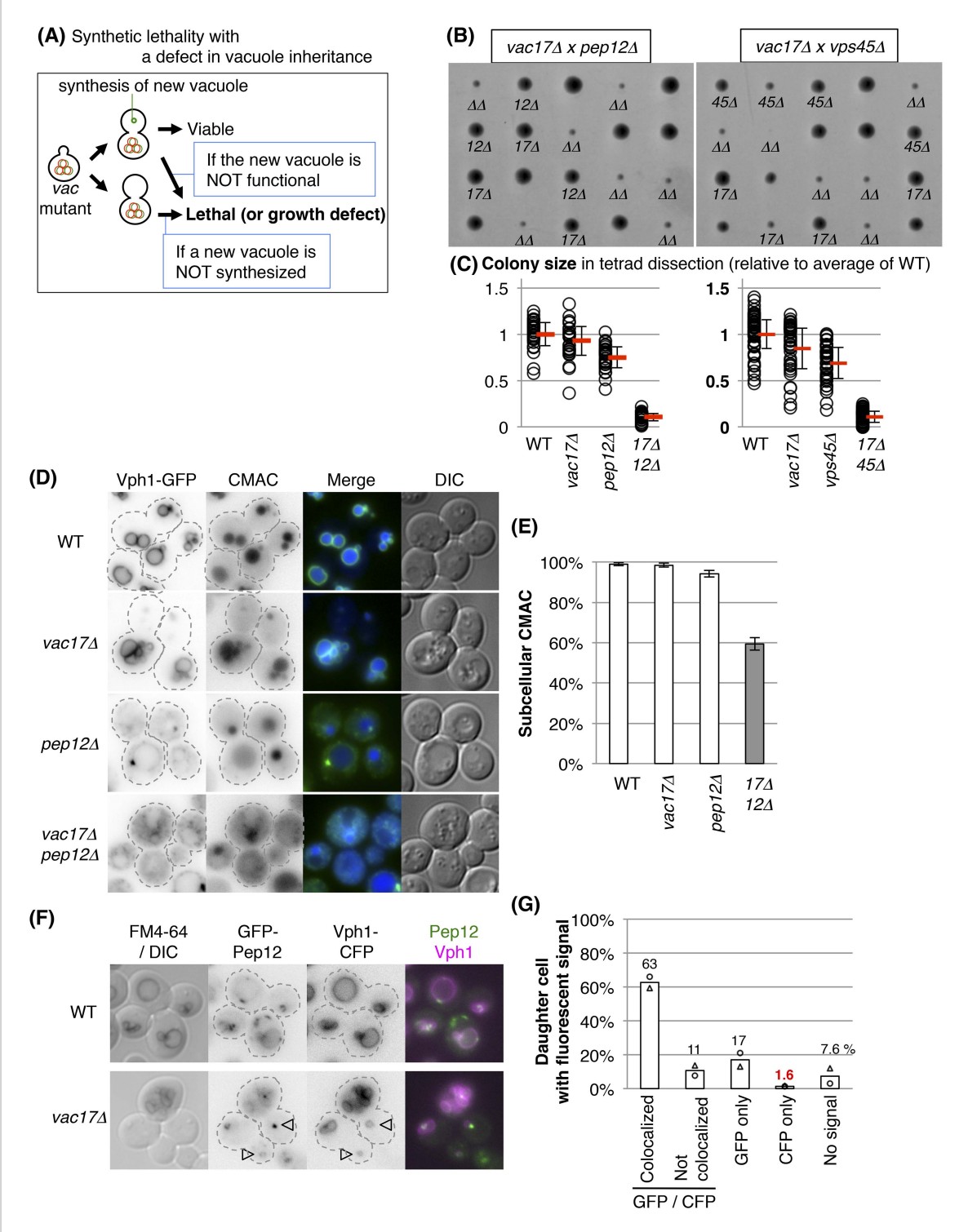

**Figure 2**. Pep12 and Vps45 are required for the synthesis of a new vacuole. (**A**) Schematic of pathways predicted to exhibit synthetic lethality with mutations in vacuole inheritance. When vacuole inheritance is defective, the bud generates a new vacuole that is independent of the mother vacuole. If vacuoles play an essential role, then cells with no mechanism to generate a vacuole will not be viable. Furthermore if the new vacuole is defective in its essential function(s), the cell will not be viable. (**B**) The *pep12Δ* and *vps45Δ* mutants exhibit a synthetic growth defect with *vac17Δ*. Results of tetrad dissection. Haploid colonies from tetrads derived from heterozygous diploids of *VAC17/vac17Δ PEP12/pep12Δ* and *VAC17/vac17Δ VPS45/vps45Δ* were arrayed vertically on YPD (rich medium) plates incubated at 24°C for 3 days. *vac17Δ = 17Δ*; *pep12Δ = 12Δ*; *vps45Δ = 45Δ*; *vac17Δ pep12Δ* or *vac17Δ vps45Δ* double mutant *= ΔΔ* are indicated. (**C**) Quantification of colony size, relative to the average of wild-type colonies. A total of 28 full tetrads and 48 full

*Figure 2. continued on next page*

Figure 2. Continued

tetrads were analyzed for *vac17Δ pep12Δ* and *vac17Δ vps45Δ*, respectively. Average size in each genotype (red bar). Error bar; SD. (**D**) Both vacuole inheritance and new synthesis are important to maintain functional vacuoles. In the *vac17Δ pep12Δ* mutant several cells appear to lack a vacuole. Wild-type cells incubated with 10 μM CMAC for 30 min exhibited a blue fluorescent signal in the vacuole lumen. The limiting membrane of the vacuole is indicated by Vph1-GFP expressed from its endogenous locus. Wild-type and *vac17Δ* cells show normal localization of Vph1-GFP and CMAC. Single *pep12Δ* cells show abnormal distribution in Vph1-GFP, but not CMAC. The *vac17Δ pep12Δ* double mutant cells show defects in the localization of Vph1-GFP and CMAC. (**E**) Quantification of cells with a CMAC positive subcellular structure. Any CMAC containing structure with or without Vph1-GFP was scored as a structure. Error bars; SD calculated from four independent experiments with at least 100 cells counted in each strain/experiment. (**F**) New vacuoles are generated from Pep12-positive endosomes. GFP-Pep12/Vph1-CFP expressed in wild-type and *vac17Δ* cells were pulse labeled with FM4-64. GFP-Pep12 and Vph1-CFP were expressed from the endogenous loci in both strains. Open arrowheads; new vacuoles. (**G**) Quantification of percent daughter cells with Vph1-CFP and/or GFP-Pep12 in *vac17Δ* cells. Averages from two independent experiments; at least 100 cells counted per experiment. Open circles and triangles indicate each experiment.

The following figure supplement is available for figure 2:

**Figure supplement 1**. Pep12 and Vps45 are required for the viability of vacuole inheritance mutants.

*PEP12* localization is consistent with its role in the synthesis of a new vacuole. GFP-Pep12 and Vph1-CFP were co-expressed in a *vac17Δ* mutant labeled with FM4-64. In this strain, 63% of *vac17Δ* cells had GFP-Pep12 on the newly synthesized vacuole in the bud, which was Vph1-CFP positive but lacked FM4-64 (**Figure 2F,G**). In 17% of daughter cells, GFP-Pep12 was present in buds without a vacuole, as indicated by the absence of Vph1-CFP (vacuole). This suggests that a Pep12-positive endosome appears first, and subsequently a Vph1-positive vacuole matures from the Pep12-positive endosome. Note that in only 1.6% of cells, Vph1-CFP was present without GFP-Pep12.

## The vacuole is required for cell-cycle progression from early G1

That the vacuole in the mother cell must reach a specific size prior to bud emergence (**Figure 1B**) and that a vacuole is required for cell growth (**Figure 2**), raised the possibility that the vacuole is required for cell-cycle progression. Thus, we tested whether the *vac17 pep12* double mutant arrests at a specific point in the cell-cycle. To perform this analysis, we used the *pep12-60^{tsf}* mutant, which is <u>t</u>emperature <u>s</u>ensitive for <u>f</u>unction (*tsf*). At elevated temperatures *PEP12* function is acutely ablated, but the cells remain viable (**Burd et al., 1997**). Importantly, the *vac17Δ pep12-60^{tsf}* double mutant, but not *pep12-60^{tsf}* single mutant, showed a severe growth defect at 37°C (**Figure 3A**).

To test cell-cycle progression, we labeled DNA with propidium iodide (PI), and measured DNA content via FACS analysis. Wild-type, *vac17Δ*, *pep12-60^{tsf}*, and *vac17Δ pep12-60^{tsf}* cells were incubated overnight at 24°C, then shifted to 37°C. At 24°C (0 hr of 37°C), the *vac17Δ pep12-60^{tsf}* double mutant had a normal cell-cycle profile (**Figure 3B** and **Figure 3—figure supplement 1A**). After incubation at 37°C for 24 hr, wild-type, *vac17Δ* and *pep12-60^{tsf}* showed a similar percent of G1 phase cells (1N DNA); 46($\pm$2)%, 48($\pm$2)%, and 43($\pm$3)%, respectively (**Figure 3B** and **Figure 3—figure supplement 1A**). In contrast, after incubation at the restrictive temperature for 24 hr, 80($\pm$3)% of the *vac17Δ pep12-60^{tsf}* double mutant cells arrested at G1 phase (1N DNA). Importantly, at 8, 12 and 24 hr after the shift to 37°C, the differences between the *vac17Δ pep12-60^{tsf}* double mutant and the other strains were statistically significant (all p-values $< 1 \times 10^{-3}$). Consistent with these findings, both the *vac17Δ pep12Δ* and *vac17Δ vps45Δ* double mutants exhibited an arrest in G1 phase (**Figure 3—figure supplements 1B,C**). These results strongly suggest that a functional vacuole is important for cell-cycle progression from G1 phase.

The finding that the *vac17Δ pep12-60^{tsf}* double mutant arrests in G1 phase, suggested that this might be the primary reason for the growth arrest (**Figure 3A**). To further test this hypothesis, we monitored cell viability at several time points following a shift to 37°C. Cells were counted with a hemocytometer, and the ability of these cells to form colonies at 24°C was assessed (**Figure 3B**). There was a measurable increase in the lethality of the *vac17Δ pep12-60^{tsf}* mutant. However the difference between the lethality of the *pep12-60^{tsf}* single mutant and the *vac17Δ pep12-60^{tsf}* double mutant was not statistically significant (p = 0.082) (**Figure 3B**). Together, these observations strongly suggest that cells without a functional vacuole arrest at G1 phase, and that this arrest is not an artifact of general cell death.

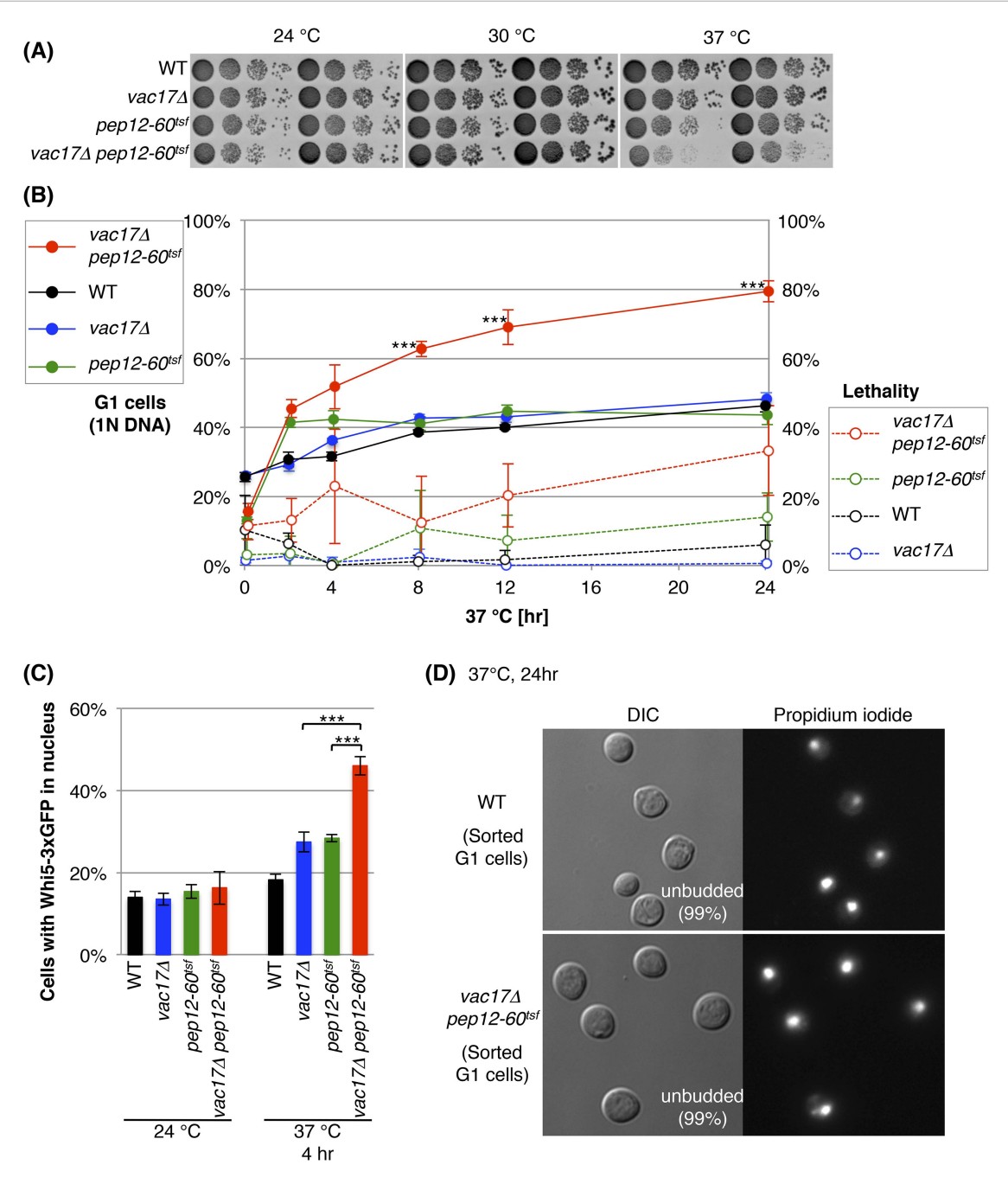

**Figure 3**. The vacuole is required for cell-cycle progression from early G1. (**A**) The *vac17Δ pep12-60^tsf* double mutant shows synthetic growth defects at the restrictive temperature, 37˚C. Wild-type, *vac17Δ*, *pep12-60^tsf* and *vac17Δ pep12-60^tsf* strains were cultured in liquid media and serial dilutions were spotted onto YPD plates. The plates were incubated at 24˚C, 30˚C and 37˚C for 2 days. (**B**) The *vac17Δ pep12-60^tsf* double mutant arrests in G1 phase at the restrictive temperature 37˚C. Percent cells in G1 phase (solid lines). Yeast strains tested; wild-type, *vac17Δ*, *pep12-60^tsf*, and *vac17Δ pep12-60^tsf*. Cultures were incubated at 24˚C overnight, and then sifted to 37˚C for 0, 2, 4, 8, 12, or 24 hr. The percentage of G1 cells (1N DNA) was measured using propidium iodide (PI) staining and assessed by flow cytometry. The same cultures were analyzed for lethality (percent dead cells) (dashed lines). After incubation at 37˚C, the number of yeast cells were assessed with a hemocytometer, and their ability to form colonies at 24˚C on YPD plates was tested. Lethality was inferred from the number of cells that survived the treatment. Error bars; SD calculated from four independent experiments. *** (p-value < 1 × 10⁻³). (**C**) The *vac17Δ pep12-60^tsf* double mutant arrests in early G1 phase at the restrictive temperature 37˚C. Cells were scored for the presence of Whi5-3xGFP in the nucleus. Wild-type, *vac17Δ*, *pep12-60^tsf*, and *vac17Δ pep12-60^tsf* cells, which express Whi5-3xGFP from its endogenous locus, were incubated at 24˚C overnight, and then sifted to 37˚C for 0 or 4 hr. Error bars; SD calculated from three independent experiments with at least 100 cells counted in each strain/experiment. *** (p-value < 1 × 10⁻³). (**D**) Arrested cells that have 1N DNA content are unbudded. Wild-type and *vac17Δ pep12-60^tsf* cells were

*Figure 3. continued on next page*

*Figure 3. Continued*

incubated at 24°C overnight, and then sifted to 37°C for 24 hr. After fixation, yeast were stained with PI, and cells with 1N DNA were sorted by flow cytometry. The sorted cells were observed by microscopy. For both wild-type and the *vac17Δ pep12-60^tsf* double mutant 99% of the cells with 1N DNA were unbudded. Sorted cells from three individual experiments were counted. At least 400 cells were counted for each experiment.

The following figure supplement is available for figure 3:

**Figure supplement 1**. The vacuole is required for cell-cycle progression from early G1 phase.

Additional evidence for a specific arrest in G1 phase, came from the finding that when the *vac17Δ pep12-60^tsf* mutant was incubated at the restrictive temperature, there was a striking increase of cells with Whi5 in the nucleus (*Figure 3C* and *Figure 3—figure supplement 1D*). Whi5 is a transcriptional repressor of the SBF complex (Swi4-Swi6), and is localized in the nucleus in early G1 phase (*Costanzo et al., 2004*; *de Bruin et al., 2004*). In wild-type cells, Whi5 nuclear localization is transient and released by Cdc28-Cln3 activity, which enables progression to early G1 phase.

Further evidence that this is a *bona fide* G1 arrest, came from the finding that the *vac17Δ pep12-60^tsf* mutant with 1N DNA content, arrests as unbudded cells. We collected 1N DNA cells by flow cytometry, and determined their morphology by microscopy. After incubation at 37°C for 24 hr, 99% of the G1 cells were unbudded, in both the wild-type and *vac17Δ pep12-60^tsf* mutant (*Figure 3D*). Together these results indicate that the vacuole is required for early G1 progression.

## TORC1-*SCH9* signaling from the new vacuole is required for cell-cycle progression

The above findings predict that regulation of the cell-cycle requires signaling from the vacuole. Evidence for a candidate signaling pathway came from studies which showed that deletion of Target Of Rapamycin 1 (*TOR1*) showed synthetic growth defects with *vac17Δ* and *vac8Δ* (*Zurita-Martinez et al., 2007*; *Costanzo et al., 2010*) (*Figure 4—fiugre supplements 1A,B*). Similarly, another vacuole inheritance mutant, *myo2-N1304D*, was synthetic lethal with *tor1Δ* (*Figure 4—figure supplement 1C*). Notably, the *vac17Δ tor1Δ* double mutant showed an increase in cells arrested at G1 phase (*Figure 4A,B*). This arrest in G1 phase was similar to that observed for the *vac17Δ pep12-60^tsf* and *vac17Δ pep12Δ* mutants (*Figure 3—figure supplements 1A–C*). This suggests that TORC1 signaling from the vacuole may account at least in part for the G1 arrest observed in the *vac17Δ pep12-60^tsf* and *vac17Δ pep12Δ* mutants. Interestingly, the *vac17Δ tor1Δ* mutants generated a new vacuole in the daughter cells (*Figure 4C*). This suggests that *TOR1* functions after the synthesis of the new vacuole.

*TOR1* encodes a PIK-related protein kinase (*Alarcon et al., 1999*). In yeast, Tor1 functions in the TORC1 complex, which is composed of Tor1/2, Kog1, Lst8, and Tco89 (*Loewith et al., 2002*). In yeast, TORC1 localizes on the vacuole membrane (*Reinke et al., 2004*; *Araki et al., 2005*; *Urban et al., 2007*; *Sturgill et al., 2008*; *Binda et al., 2009*; *Jin et al., 2014*), and is a key determinant of nutrient status (*Di Como and Arndt, 1996*). We found that the kinase activity of Tor1 was required for growth of *vac17Δ* (*Figure 4D*), and that the TORC1 specific subunit *KOG1* was also required for growth of *vac17Δ* (*Figure 4—figure supplement 1D*). In addition, the *lst8-15* temperature sensitive mutant is also synthetically lethal with *vac17Δ* (*Costanzo et al., 2010*). These results suggest that the kinase activity of the TORC1 complex is required for normal growth of vacuole inheritance mutants.

One critical target of TORC1 is the Sch9 kinase, which shares overlapping functions with metazoan S6 kinase (*Ballou et al., 1991*; *Oldham et al., 2000*; *Christie et al., 2002*). TORC1 directly phosphorylates Sch9 on several serines and threonines, and the phospho-mimetic *sch9-2D3E*, but not wild-type *SCH9* or the Ala-substituted *sch9-5A* mutant bypasses the TORC1 inhibitor, rapamycin (*Loewith et al., 2002*; *Urban et al., 2007*). Notably, the phospho-mimetic *sch9-2D3E* mutant partially suppressed the growth defect of the *vac17Δ tor1Δ* mutant (*Figure 4E*), suggesting that the arrest of this mutant is due in part to defects in TORC1 mediated signaling via *SCH9*.

It was previously shown that Sch9-2D3E localizes on the vacuole membrane (*Urban et al., 2007*). To test whether bypass of Tor1 by *sch9-2D3E* requires a functional vacuole, we tested whether *sch9-2D3E* can suppress the *vac17Δ pep12Δ* double mutant. Notably *sch9-2D3E* did not suppress the *vac17Δ pep12Δ* double mutant (*Figure 4F*). This suggests that a functional vacuole is required for the roles of the TORC1-*SCH9* pathway in cell-cycle progression from G1 phase.

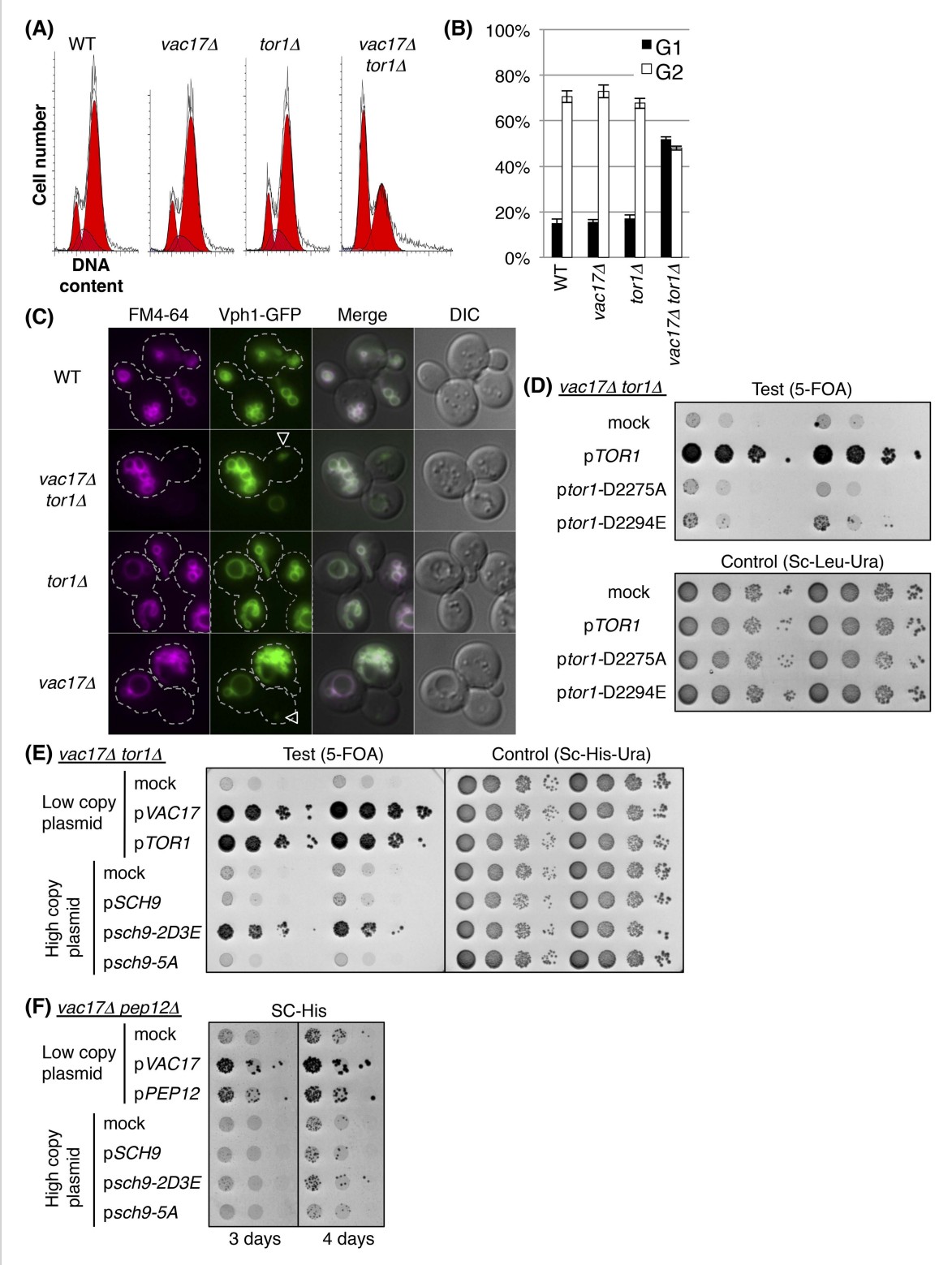

**Figure 4**. TORC1-*SCH9* signaling from the new vacuole is required for cell-cycle progression. (**A**) The *vac17Δ tor1Δ* double mutant exhibits an accumulation of G1 phase cells. Flow cytometry analysis with PI staining of yeast strains; wild-type, *vac17Δ*, *tor1Δ*, and *vac17Δ tor1Δ*. (**B**) Quantification of percent cells in G1 and G2 phase. Error bars; SD calculated from four independent experiments. (**C**) A new vacuole is synthesized in the new daughter cells of the *vac17Δ tor1Δ* double mutant. Wild-type, *vac17Δ*, *tor1Δ*, and *vac17Δ tor1Δ* cells which express Vph1-GFP from its endogenous locus, were pulse labeled with FM4-64. Arrowheads; new vacuole in daughter cells. (**D**) The kinase activity of target of rapamycin 1 (Tor1) is required for growth of the

*Figure 4. continued on next page*

*Figure 4. Continued*

vacuole inheritance mutant, *vac17Δ*. Plasmids were transformed into a *vac17Δ tor1Δ* mutant containing pRS416 [*URA3*] *TOR1*. Plasmids tested were pRS315 [*LEU2*] (mock), pRS315 *HA-TOR1*, pRS315 *HA-tor1-D2275A*, or pRS315 *HA-tor1-D2294E*. Transformed colonies were cultured in liquid media and serial dilutions spotted onto SC+5-FOA or SC-Leu-Ura plates. Plates were incubated at 24°C for 4 days. (**E**) TORC1 signals from the new vacuole via Sch9. The phospho-mimetic *sch9-2D3E* mutant partially rescues the growth defect of the *vac17Δ tor1Δ* mutant. pRS413 (mock), pRS413 *VAC17*, pRS413 *TOR1*, pVT102-H (mock), pVT102-H *SCH9*, pVT102-H *sch9-2D3E*, or pVT102-H *sch9-5A* expressed in *vac17Δ tor1Δ* with pRS416 *TOR1*. Transformed colonies were cultured in liquid media and serial dilutions were spotted onto SC-His+5-FOA or SC-His-Ura plates, and incubated at 24°C for 4 days. (**F**) Sch9 signaling requires a functional vacuole. The phospho-mimetic *sch9-2D3E* mutant does not rescue the growth defect of the *vac17Δ pep12Δ* mutant. pRS413 (mock), pRS413 *VAC17*, pRS413 *TOR1*, pVT102-H (mock), pVT102-H *SCH9*, pVT102-H *sch9-2D3E*, or pVT102-H *sch9-5A* plasmids were expressed in a *vac17Δ pep12Δ* strain. Transformed colonies were cultured in liquid media and serial dilutions spotted onto an SC-His plate, and incubated at 24°C for 3 to 4 days.

The following figure supplement is available for figure 4:

**Figure supplement 1**. TORC1-*SCH9* is required for the viability of vacuole inheritance mutants.

While previous studies showed that TORC1 signals from the vacuole/lysosome (*Sancak et al., 2010*), and that Sch9 is activated in that location (*Urban et al., 2007*), it was assumed that once Sch9 is activated, it no longer requires the vacuole for its further downstream functions. However our findings strongly suggest that the vacuole is required for Sch9 function(s) after Sch9 is phosphorylated by TORC1. One possible role of the vacuole in Sch9 function, is that target protein(s) of the Sch9 kinase must be present on the vacuole membrane. Alternatively or in addition, the full kinase activity of Sch9 may require other proteins that are on the vacuole membrane.

## The newly synthesized vacuoles initially lack Sch9 and Fab1

If a cell does not receive a vacuole from the mother cell, the daughter cell generates a new vacuole. The observation that these new vacuoles grow to a specific size prior to generation of a bud, and that TORC1 signaling is also required, raised the possibility that that there are functional differences between newly synthesized vacuoles and inherited vacuoles. As a first approach, we tested the localization of several proteins that are involved in the TORC1 pathway, Tor1, Kog1 and Sch9. New vacuoles were defined as Vph1-CFP positive structures that failed to inherit FM4-64. Notably, in *vac17Δ* cells, GFP-Sch9 was defective in its localization to the new vacuoles, while the localization of Tor1 and Kog1 were unaffected (*Figure 5A,B*). To directly address whether Sch9 is eventually recruited to the newly formed vacuole and when this occurs, we correlated the presence of fluorescent signals for Vph1-CFP, Tor1-3xGFP and GFP-Sch9 (*Figure 5C,D*). These analyses show that Sch9 recruitment to the newly formed vacuole is slower than the recruitment of Vph1 and Tor1. Specifically, small budded cells did not have fluorescent signals for any of the proteins, which indicates that these small buds do not have a vacuole (*Figure 5C–F*). As the bud increases in size, in most cases, Vph1-CFP and Tor1-3xGFP appeared simultaneously (*Figure 5C*). This indicates that Tor1-3xGFP is immediately recruited to the newly formed vacuoles. Moreover in some small budded cells, Tor1-3xGFP was present without Vph1-CFP, which suggests that Tor1 may be present at endosomes in these small budded cells.

In the *vac17Δ* mutant, although Tor1-3xGFP and Vph1-CFP are both present on the new vacuoles in medium buds (approximately 0.6 daughter size/mother size) (*Figure 5C*), GFP-Sch9 was generally not present until the bud size was larger (approximately 0.8) (*Figure 5D*). Thus Sch9 recruitment is delayed compared to Vph1 and Tor1, but eventually occurs. Notably the recruitment of the Sch9-2D3E mutant was similar to the recruitment of wild-type Sch9 (*Figure 5E*), indicating that the growth suppression of the *vac17Δ tor1Δ* mutant by *sch9-2D3E* is not due to a faster recruitment to the vacuole.

$PI(3,5)P_2$, an inositol lipid which is present at the vacuole membrane, is required for vacuole membrane targeting of Sch9 (*Jin et al., 2014*). To test whether the delay in Sch9 recruitment might be due to a delay in the generation of $PI(3,5)P_2$ on newly formed vacuoles, we tested the timing of the recruitment of Fab1, the sole lipid kinase that generates $PI(3,5)P_2$ (*Gary et al., 1998*). Notably the timing of the recruitment of Fab1 was similar to that observed for Sch9. In a *vac17Δ* mutant which co-expressed Vph1-CFP and Fab1-3xGFP, only Vph1-CFP is present in medium buds (approximately 0.6 daughter size/mother size) (*Figure 5F*). The average bud size where both Vph1 and Fab1 are present was 0.76(±0.10). Thus Fab1 recruitment is delayed compared to Tor1, but is similar to that observed for

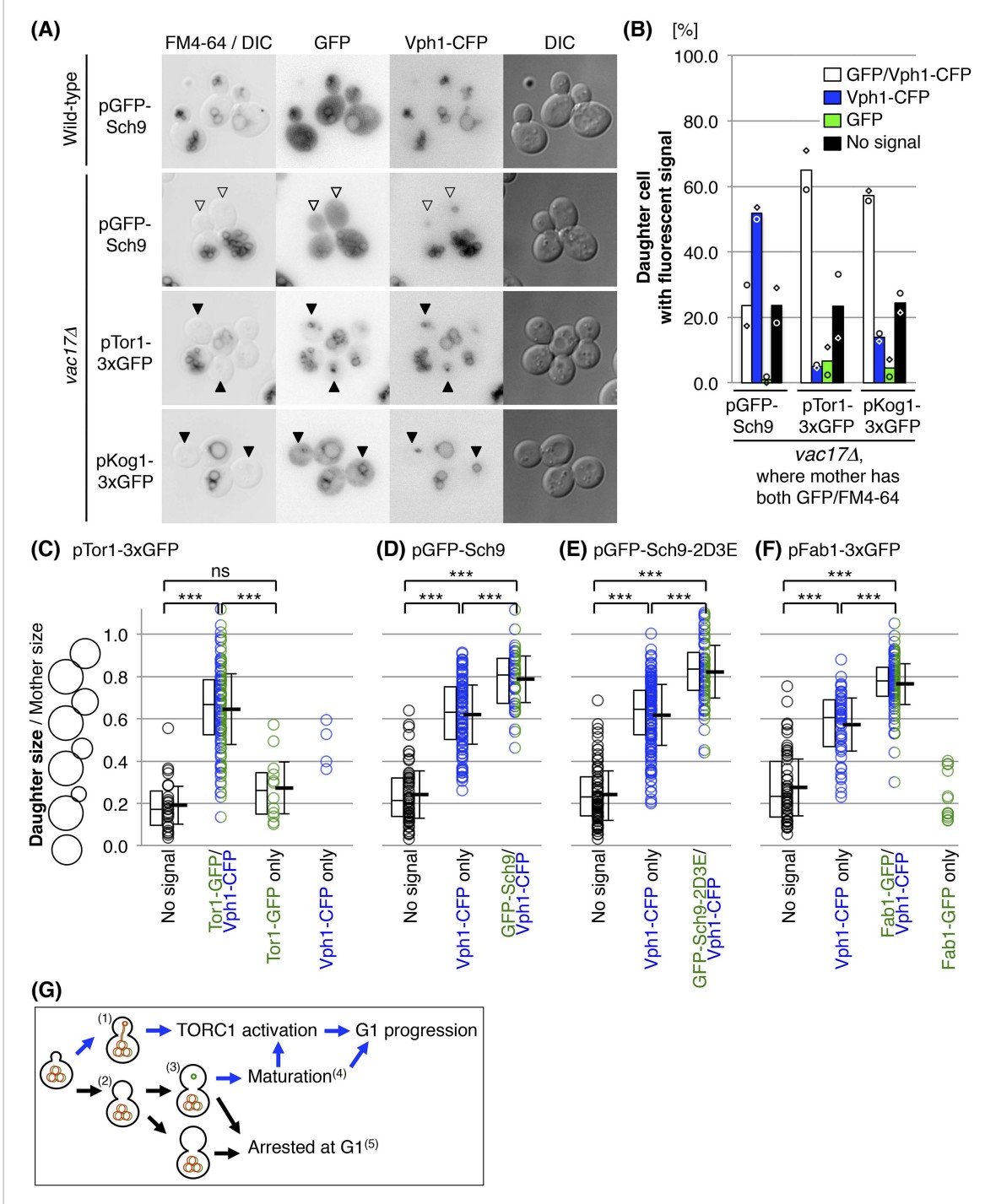

**Figure 5**. The newly synthesized vacuoles initially lack Sch9 and Fab1. (**A**) Sch9 does not localize to the newly formed vacuole. Indicated plasmids were transformed into wild-type and *vac17Δ* strains, which express Vph1-CFP from its endogenous locus: pRS416 *GFP-SCH9*, pRS416 *TOR1-3xGFP*[(D330)], or pRS416 *KOG1-3xGFP*. Transformed cells were pulse labeled with FM4-64. Open arrowheads indicate a newly formed vacuole (Vph1-CFP) that was not inherited (lack of FM4-64), and is lacking GFP-Sch9. Closed arrowheads indicate a newly formed vacuole (Vph1-CFP) that was not inherited (lack of FM4-64), and with the GFP fusion protein, either Tor1-3xGFP[(D330)] or Kog1-3xGFP. (**B**) Quantification of cells with fluorescence (Vph1-CFP and/or GFP) in daughter cells, where the mother has both GFP and FM4-64 signals. Averages from two independent experiments, with n = 69 and n = 104 for GFP-Sch9, n = 166 and n = 110 for Tor1-3xGFP[(D330)], and n = 106 and n = 126 cells for Kog1-3xGFP, respectively. Open circles and squares indicate results of each experiment. (**C**) Tor1 is immediately recruited to the newly formed vacuoles. FM4-64 labeled *vac17Δ* cells that express Vph1-CFP from its endogenous locus, and Tor1-3xGFP expressed from a CEN plasmid with its endogenous promoter were used. Most small budded cells do not have fluorescent signals

*Figure 5. continued on next page*

*Figure 5. Continued*

for any of the proteins, indicating that these small buds do not have a vacuole. In most cases, as the bud increases in size, Vph1-CFP and Tor1-3xGFP appear simultaneously. The middle line in the box plot indicates the median of the data set. The upper edge of the box indicates the 75th percentile of the data set, and the lower edge indicates the 25th percentile. ns; not a significant difference (p-value > 0.10); *** (p-value < $1 \times 10^{-6}$). (**D**) Sch9 recruitment is delayed compared to Tor1, but eventually occurs. FM4-64 labeled *vac17Δ* cells expresses Vph1-CFP from its endogenous locus, and GFP-Sch9 expressed from a CEN plasmid with its endogenous promoter were used. In medium sized buds (0.62(±0.14) daughter size/mother size) only Vph1-CFP is present. The average bud size where both Vph1 and Sch9 are present is 0.79(±0.11). (**E**) Recruitment of Sch9-2D3E to the new vacuole is similar to the recruitment of wild-type Sch9. FM4-64 labeled *vac17Δ* cells which express Vph1-CFP from its endogenous locus, and GFP-Sch9-2D3E expressed from a CEN plasmid with its endogenous promoter were used. (**F**) The timing of the recruitment of Fab1 was similar to that observed for Sch9. FM4-64 labeled *vac17Δ* cells which express Vph1-CFP from its endogenous locus, and Fab1-3xGFP expressed from a CEN plasmid with its endogenous promoter were used. In medium sized buds (0.57(±0.13) daughter size/mother size) only Vph1-CFP is present. The average bud size where both Vph1 and Fab1 are present is 0.76(±0.10). (**G**) Model: The vacuole is essential for cell-cycle progression and functions in part through the TORC1 pathway. When the daughter cell receives vacuoles from the mother cell[1], the daughter can progress from G1. If the cell fails to inherit a vacuole[2], the cell generates a new vacuole[3], which is followed by maturation of the new vacuole prior to G1 progression[4]. Without a functional vacuole, the daughter cell arrests at G1 phase[5].

Sch9. These observations suggest that there are differences between new vacuoles and inherited vacuoles, and that the newly synthesized vacuoles are missing specific components of mature vacuoles that are essential for cell-cycle progression.

Together, these observations demonstrate that a functional vacuole is crucial for cell-cycle progression at G1 phase, and that the TORC1-*SCH9* pathway is part of this critical function (*Figure 4G*). TORC1-*SCH9* signaling from the vacuole may be involved in G1 progression through its known functions in a ribosome biogenesis and translation (*Barbet et al., 1996*; *Jorgensen et al., 2004*; *Urban et al., 2007*). Alternatively, the TORC1-*SCH9* pathway may signal the cell-cycle machinery that a functional vacuole is present and hence the cell is ready to progress from G1. Previous studies showed that Sch9 activity is critical for cell size (*Jorgensen et al., 2004*; *Urban et al., 2007*), and that cell size correlates with vacuole size (*Chan and Marshall, 2014*). These observations together with the current study suggest that TORC1-*SCH9* localization and signaling from the vacuole is critical for the regulation of cell size. It is tempting to speculate that an analogous regulation of the cell-cycle occurs from endo-lysosomal membranes in other organisms. In addition, these findings lead to the hypothesis that cells possess novel checkpoint mechanisms that prevent cell-cycle progression at G1 phase in the absence of essential organelles.

## Materials and methods

### Yeast strain and media

Yeast strains used are in *Table 1*. Deletion and fusion strains were constructed as described (*Longtine et al., 1998*). A *vac17Δ pep12-60^tsf* double mutant strain was made through mating *pep12-60^tsf* (CBY9) (*Burd et al., 1997*) with *vac17Δ* (LWY5798) (*Ishikawa et al., 2003*). To generate a *GFP-PEP12::natNT2* strain, a *ClaI-ApaI* fragment from pBlueScript SK+ (pBS) *GFP-PEP12::natNT2* vector was integrated into the *PEP12* locus. Yeast cultures were grown at 24°C unless stated otherwise. Yeast extract-peptone-dextrose (1% yeast extract, 2% peptone, 2% dextrose; YEPD), synthetic complete (SC) lacking the appropriate supplement(s), and 5-FOA media were made as described (*Kaiser et al., 1994*). Unless stated otherwise, SC medium contained 2% dextrose.

### In vivo labeling of vacuoles

Vacuoles were labeled in vivo with N-(3-triethelammoniumpropyl)-4-(6 (4-(diethylamino) phenyl) hexatrienyl) pyridinium dibromide (FM4-64 [SynaptoRed C2]; Biotium, Hayward, CA, United States) essentially as described (*Ishikawa et al., 2003*). In brief, a 2 mg/ml stock solution of FM4-64 in dimethyl sulfoxide was added to early log phase cultures for a final concentration of 80 µM. After 1 hr of labeling, cells were washed and then chased in fresh liquid medium for 3–4 hr.

### Plasmids

Plasmids used are in *Table 2*. To generate an integration vector to express GFP fused to Pep12 from the *PEP12* gene locus, pBS *GFP-PEP12::natNT2* was made. A 1.4 kb *ClaI-BstBI* fragment of *PEP12* was inserted at the *ClaI* site of pBS. To insert GFP at the N-terminus of Pep12, an *AvrII* site was generated

**Table 1**. Yeast strains used in this study

| Strain | Genotype | Source | Figure |
|---|---|---|---|
| LWY7235 | MATa, ura3-52, leu2-3,-112, his3-Δ200, trp1-Δ901, lys2-801, suc2-Δ9 | (*Bonangelino et al., 1997*) | – |
| LWY11678 | MATa, VPH1-GFP::KanMX | This study | *Figures 1, 2, 4, Figure 1—figure supplement 1* |
| LWY12144 | MATa, VPH1-GFP::KanMX, vac17Δ::TRP1 | This study | *Figures 1, 2, 4, Figure 1—figure supplement 1* |
| LWY15258 | MATa/α, VAC17/vac17Δ::TRP1, PEP12/pep12Δ::KanMX | This study | *Figure 2* |
| LWY15612 | MATa/α, VAC17/vac17Δ::TRP1, VPS45/vps45Δ::KanMX | This study | *Figure 2* |
| LWY14490 | MATa, VPH1-GFP::KanMX, pep12Δ::KanMX | This study | *Figure 2* |
| LWY14493 | MATa, VPH1-GFP::KanMX, vac17Δ::TRP1, pep12Δ::KanMX | This study | *Figure 2* |
| LWY15515 | MATa, GFP-PEP12::natNT2, VPH1-CFP::KanMX | This study | *Figure 2* |
| LWY15506 | MATa, GFP-PEP12::natNT2, VPH1-CFP::KanMX, vac17Δ::TRP1 | This study | *Figure 2* |
| LWY15263, LWY14462, LWY12369 | MATa | This study | *Figures 3, 4, Figure 3—figure supplement 1* |
| LWY5798 | MATa, vac17Δ::TRP1 | (*Tang et al., 2003*) | – |
| LWY15244, LWY14468, LWY12366 | MATa, vac17Δ::TRP1 | This study | *Figures 3, 4, Figure 3—figure supplement 1* |
| CBY9 | MATα, pep12-60$^{tsf}$, leu2-3,112::pBHY11 CPY-Inv LEU2 | (*Burd et al., 1997*) | – |
| LWY15250 | MATa, pep12-60$^{tsf}$ | This study | *Figure 3, Figure 3—figure supplement 1* |
| LWY15249 | MATa, vac17Δ::TRP1, pep12-60$^{tsf}$ | This study | *Figure 3, Figure 3—figure supplement 1* |
| LWY15799 | MATa, WHI5-3xGFP::His3MX | This study | *Figure 3, Figure 3—figure supplement 1* |
| LWY15791 | MATa, WHI5-3xGFP::His3MX, vac17Δ::TRP1 | This study | *Figure 3, Figure 3—figure supplement 1* |
| LWY15789 | MATa, WHI5-3xGFP::His3MX, pep12-60$^{tsf}$ | This study | *Figure 3, Figure 3—figure supplement 1* |
| LWY15814 | MATα, WHI5-3xGFP::His3MX, vac17Δ::TRP1, pep12-60$^{tsf}$ | This study | *Figure 3, Figure 3—figure supplement 1* |
| LWY12364 | MATa, tor1Δ::KanMX | This study | *Figure 4* |
| LWY12367 | MATα, vac17Δ::TRP1, tor1Δ::KanMX | This study | *Figure 4* |
| LWY12168 | MATa, VPH1-GFP::KanMX, tor1Δ::KanMX | This study | *Figure 4* |
| LWY12193 | MATa, VPH1-GFP::KanMX, vac17Δ::TRP1, tor1Δ::KanMX | This study | *Figure 4* |
| LWY14142 | MATa, vac17Δ::TRP1, tor1Δ::KanMX, pRS416 TOR1 | This study | *Figure 4* |
| LWY12358, LWY14487 | MATa, vac17Δ::TRP1, pep12Δ::KanMX | This study | *Figure 4, Figure 3—figure supplement 1* |
| LWY11657 | MATa, VPH1-CFP::KanMX | This study | *Figure 5* |
| LWY13781 | MATa, VPH1-CFP::KanMX, vac17Δ::TRP1 | This study | *Figure 5* |
| LWY15610 | MATa/α, VAC8/vac8Δ::HIS3, PEP12/pep12Δ::KanMX | This study | *Figure 2—figure supplement 1* |
| LWY15614 | MATa/α, VAC8/vac8Δ::HIS3, VPS45/vps45Δ::KanMX | This study | *Figure 2—figure supplement 1* |

*Table 1. Continued on next page*

Table 1. Continued

| Strain | Genotype | Source | Figure |
|---|---|---|---|
| LWY2947 | MATα, myo2Δ::TRP1, YCp50-MYO2 | (Catlett and Weisman, 1998) | Figure 2—figure supplement 1, Figure 4—figure supplement 1 |
| LWY12443 | MATα, pep12Δ::KanMX, myo2Δ::TRP1, YCp50-MYO2 | This study | Figure 2—figure supplement 1 |
| LWY15581 | MATα, vps45Δ::KanMX, myo2Δ::TRP1, YCp50-MYO2 | This study | Figure 2—figure supplement 1 |
| LWY14497 | MATa, pep12Δ::KanMX | This study | Figure 3—figure supplement 1 |
| LWY14475 | MATa, vps45Δ::KanMX | This study | Figure 3—figure supplement 1 |
| LWY14463 | MATa, vac17Δ::TRP1, vps45Δ::KanMX | This study | Figure 3—figure supplement 1 |
| LWY1 | MATa/α, TOR1/tor1Δ::KanMX | This study | Figure 4—figure supplement 1 |
| LWY15616 | MATa/α, VAC8/vac8Δ::HIS3, TOR1/tor1Δ::KanMX | This study | Figure 4—figure supplement 1 |
| LWY12001 | MATa, tor1Δ::KanMX, myo2Δ::TRP1, YCp50-MYO2 | This study | Figure 4—figure supplement 1 |
| LWY13595 | MATa, vac17Δ::TRP1, kog1Δ::KanMX, pRS316 KOG1 | This study | Figure 4—figure supplement 1 |

Each above haploid strain is ura3-52, leu2-3,-112, his3-Δ200, trp1-Δ901, lys2-801, suc2-Δ9, and diploid strain is ura3-52/ura3-52, leu2-3,-112/leu2-3,-112, his3-Δ200/his3-Δ200, trp1-Δ901/trp1-Δ901, lys2-801/lys2-801, suc2-Δ9/suc2-Δ9.

at the N-terminus of *PEP12* by PCR using primers (5′-CAA TAA TTG TGT TGA GAT Gcc tag gTC GGA AGA CGA ATT TTT TGG-3′) and (5′-CCA AAA AAT TCG TCT TCC GAc cta ggC ATC TCA ACA CAA TTA TTG-3′). The GFP fragment was amplified from pFA6a GFP(S65T)-KanMX (*Longtine et al., 1998*) by PCR using primers (5′-TGA gct agc AGT AAA GGA GAA GAA CTT TTC ACT GG-3′) and (5′-TGA act agt gtt aat taa ccc ggg gat ccg tcg acc TTT GTA TAG TTC ATC CAT GCC-3′). The *Nhe*I-*Spe*I fragment of GFP was inserted at the *Avr*II site. The *natNT2* maker was amplified from pFA6a *natNT2* (*Janke et al., 2004*) by PCR using primers (5′-CTG tgt aca CAG CGA CAT GGA GGC-3′) and (5′-TCA tgt aca ACA GGT GTT GTC CTC TGA G-3′). A *Bsr*GI fragment of *natNT2* was inserted into the *Bsr*GI site at 3′ UTR of the *PEP12*.

pRS416 *TOR1* includes 227 bp upstream and 944 bp downstream of the *TOR1* gene, the same region as pRS315 *HA-TOR1* (gift from Dr Robbie Loewith).

For generation of *tor1-D2275A*, and *-D2294E* kinase dead mutants (*Zheng et al., 1995*; *Alarcon et al., 1999*), the *TOR1* gene was mutagenized by site-directed mutagenesis using the following primers: (D2275A-S) 5′-GTT ATA TTC TGG GAC TAG GTG cTC GCC ATC CAA GCA ACC TG-3′, (D2275A-AS) 5′-CAG GTT GCT TGG ATG GCG AgC ACC TAG TCC CAG AAT ATA AC-3′, (D2294E-S) 5′-CAC CGG TAA AGT TAT CCA CAT TGA aTT CGG CGA TTG TTT TGA AGC-3′, (D2294E-AS) 5′-GCT TCA AAA CAA TCG CCG AAt TCA ATG TGG ATA ACT TTA CCG GTG-3′.

For generation of pVT102-H *SCH9*, *SCH9* was amplified by PCR using primers (5′-ATA gga tcc ATG ATG AAT TTT TTT ACA TCA AAA TCG-3′) and (5′-GAG tct aga TAT TTC GAA TCT TCC ACT GAC AAA TTC-3′). A *Bam*HI-*Xba*I fragment of *SCH9* was inserted into the *Bam*HI, *Xba*I sites of pVT102-H (*Vernet et al., 1987*).

For generation of phospho-mimetic *sch9-2D3E* and non-phospho *sch9-5A* mutant (*Urban et al., 2007*), the *SCH9* gene was mutagenized by site-directed mutagenesis using the following primers: (T723D/S726D-S) 5′-CC GAT GAT GAC TGC Tga CCC GCT Aga TCC AGC CAT GCA AGC AAA G-3′, (T723D/S726D-AS) 5′-CTT TGC TTG CAT GGC TGG Atc TAG CGG Gtc AGC AGT CAT CAT CGG-3′, (T737E–S) 5′-CAA GCA AAG TTT GCT GGT TTC gaa TTT GTT GAT GAG TCC GCC ATC-3′, (T737E-AS) 5′-GAT GGC GGA CTC ATC AAC AAA ttc GAA ACC AGC AAA CTT TGC TTG-3′, (S758E/S765E–S) 5′-CCT ACA AAA Cga GTA CTT TAT GGA ACC TGG Tga aTT TAT CCC GGG-3′, (S758E/S765E-AS) 5′-CCC GGG ATA AAt tcA CCA GGT TCC ATA AAG TAC tcG TTT TGT AGG-3′, (T723A/S726A-S) 5′-CCG ATG ATG ACT GCT gCC CCG CTA gCT CCA GCC ATG CAA GCA AAG-3′, (T723A/S726A-AS) 5′-CTT TGC TTG CAT GGC TGG AGc TAG CGG GGc AGC AGT CAT CAT CGG-3′, (T737A-S) 5′-CAA

**Table 2**. Plasmids used in this study

| Plasmid name | Description | Source | Figure |
|---|---|---|---|
| pBlueScript SK+ *GFP-PEP12::natNT2* | Amp | This study | *Figure 2* |
| pRS416 *TOR1* | CEN, *URA3* | This study | *Figure 4* |
| pRS413 | CEN, *HIS3* | (*Sikorski and Hieter, 1989*) | *Figure 4* |
| pRS315 *HA-TOR1* | CEN, *HIS3* | Gift from Dr Robbie Loewith | *Figure 4* |
| pRS315 *HA-tor1-D2275A* | CEN, *HIS3* | This study | *Figure 4* |
| pRS315 *HA-tor1-D2294E* | CEN, *HIS3* | This study | *Figure 4* |
| pRS413 *VAC17* | CEN, *HIS3* | This study | *Figure 4* |
| pRS413 *TOR1* | CEN, *HIS3* | This study | *Figure 4* |
| pVT102-H | 2μ, *HIS3* | (*Vernet et al., 1987*) | *Figure 4* |
| pVT102-H *SCH9* | 2μ, *HIS3* | This study | *Figure 4* |
| pVT102-H *sch9-2D3E* | 2μ, *HIS3* | This study | *Figure 4* |
| pVT102-H *sch9-5A* | 2μ, *HIS3* | This study | *Figure 4* |
| pRS416 *GFP-SCH9* | CEN, *URA3* | (*Urban et al., 2007*) | *Figure 5* |
| pRS416 *GFP-sch9-2D3E* | CEN, *URA3* | This study | *Figure 5* |
| pRS416 *TOR1-3xGFP(D330)* | CEN, *URA3* | This study | *Figure 5* |
| pRS416 *KOG1-3xGFP* | CEN, *URA3* | This study | *Figure 5* |
| pRS416 *FAB1-3xGFP* | CEN, *URA3* | (*Jin et al., 2008*) | *Figure 5* |
| pRS413 | CEN, *HIS3* | (*Sikorski and Hieter, 1989*) | *Figure 2—figure supplement 1*, *Figure 4—figure supplement 1* |
| pRS413 *MYO2* | CEN, *HIS3* | (*Catlett and Weisman, 1998*) | *Figure 2—figure supplement 1*, *Figure 4—figure supplement 1* |
| pRS413 *myo2-N1304D* | CEN, *HIS3* | (*Catlett et al., 2000*) | *Figure 2—figure supplement 1*, *Figure 4—figure supplement 1* |
| pRS416 *KOG1* | CEN, *URA3* | (*Jin et al., 2014*) | *Figure 4—figure supplement 1* |
| pRS313 | CEN, *HIS3* | (*Sikorski and Hieter, 1989*) | *Figure 4—figure supplement 1* |
| pRS415 | CEN, *LEU2* | (*Sikorski and Hieter, 1989*) | *Figure 4—figure supplement 1* |
| pRS415 *VAC17* | CEN, *LEU2* | (*Jin et al., 2009*) | *Figure 4—figure supplement 1* |
| pRS313 *KOG1* | CEN, *HIS3* | (*Nakashima et al., 2008*) | *Figure 4—figure supplement 1* |
| pRS313 *kog1-105* | CEN, *HIS3* | (*Nakashima et al., 2008*) | *Figure 4—figure supplement 1* |

GCA AAG TTT GCT GGT TTC gCC TTT GTT GAT GAG TCC GCC ATC-3′, (T737A-AS) 5′-GAT GGC GGA CTC ATC AAC AAA GGc GAA ACC AGC AAA CTT TGC TTG-3′, (S758A/S765A-S) 5′-CCT ACA AAA CgC GTA CTT TAT GGA ACC TGG TgC CTT TAT CCC GGG-3′, (S758A/S765A-AS) 5′-CCC GGG ATA AAG GcA CCA GGT TCC ATA AAG TAC GcG TTT TGT AGG-3′.

For generation of pRS416 *TOR1-3xGFP(D330)*, an *Xba*I site was generated at the D330 position of *TOR1* with PCR using primers (5′-GTT TAT AAG GAA ATC TTG TTT TTG AAG tct Aga CCC TTT TTG AAT CAA GTG TTC-3′) and (5′-GAA CAC TTG ATT CAA AAA GGG tcT aga CTT CAA AAA CAA GAT TTC CTT ATA AAC-3′). A 3xGFP fragment was amplified from pFA6a 3xGFP-TRP1 by PCR using primers (5′-CGG tct aga GGG TTA ATT AAC GTG AGC AAG GG-3′) and (5′-AAT CTC GAG gct agc GGG GAT CCG TCG ACC CTT GTA CAG CTC GTC CAT GC-3′). An *Xba*I-*Nhe*I fragment of 3xGFP was inserted at the *Xba*I site (*Jin et al., 2014*).

For generation of pRS416 *KOG1-3xGFP*, an *Xba*I site was generated at the C-terminal end of *KOG1* by PCR using primers (5′-GAG AAT TGA TTA TTT Ttc tag aTA TGT GCC ATT TCT TTT TTT TTC-3′) and (5′-GAA AAA AAA AGA AAT GGC ACA TAt cta gaA AAA TAA TCA ATT CTC-3′). The 3xGFP fragment was amplified by PCR using primers (5′-TCT AGA GGG TTA ATT tct aga AGC AAG GGC GAG GAG C-3′) and (5′-AAT CTC GAG gct agc GTT AAT TAA CCC GGG GAT CCG TCG ACC-3′). The *Xba*I-*Nhe*I fragment of 3xGFP was inserted at the *Xba*I site (*Jin et al., 2014*).

## Flow cytometry analysis

Quantitation of nuclear DNA was determined as follows: Cells were stained with PI and analyzed by FACS analysis (MACSQuant 1; Miltenyi Biotec, Germany). In most experiments, 10,000 cells were examined. Yeast were incubated at 24°C overnight, and then sifted to 37°C for 0, 2, 4, 8, 12, or 24 hr. At the start of the experiment, yeast were in log phase growth. 1.0 $OD_{600}$ yeast cultures were collected, washed with 50 mM of Tris-HCl [pH7.5], and fixed with 70% EtOH. Cells were then washed twice with 50 mM of Tris-HCl [pH7.5], followed by sonication. Cells were treated with RNaseA (Sigma–Aldrich R6513; final 2 mg/ml in 50 mM of Tris-HCl [pH7.5]) at 37°C overnight. Cells were then treated with Pepsin (Sigma–Aldrich 7000; final 5 mg/ml) at room temperature for 30 min, and stained with PI (Sigma–Aldrich 4170) 50 mg/ml in 180 mM Tris-HCl [pH7.5], 180 mM NaCl, 70 mM $MgCl_2$ for 1 hr at room temperature. The PI stained cells were analyzed by FACS.

## Acknowledgements

We thank Drs Robbie Loewith (U of Geneva, Switzerland), Yoshiaki Kamada (National Institute for Basic Biology, Japan), and Scott Emr (Cornell University) for plasmids and yeasts. We thank Drs Yukiko Yamashita, John Kim (U of Michigan) and Bethany Strunk for helpful suggestions for the manuscript. We thank all members of the Weisman lab, especially Dr Natsuko Jin for help with the viability analyses and for discussions. We thank Emily Kauffman for technical assistance, and Steven Merz for measuring cell and vacuole diameters. We thank Dr Amy Ikui (Brooklyn College, CUNY) and the Flow Cytometry Core at the University of Michigan for FACS analysis. This work was supported by National Institutes of Health grant R37 GM062261 to LSW.

## Additional information

### Funding

| Funder | Grant reference | Author |
| --- | --- | --- |
| National Institute of General Medical Sciences (NIGMS) | R37-R01GM062261 | Lois S Weisman |

The funder had no role in study design, data collection and interpretation, or the decision to submit the work for publication.

### Author contributions

YJ, Conception and design, Acquisition of data, Analysis and interpretation of data, Drafting or revising the article; LSW, Conception and design, Analysis and interpretation of data, Drafting or revising the article

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
