## [Decision Letter]

Thank you for submitting your work entitled “The vacuole/lysosome is required for cell-cycle progression” for peer review at *eLife*. Your submission has been favorably evaluated by Vivek Malhotra (Senior Editor) and three reviewers, one of whom is a member of our Board of Reviewing Editors.

The reviewers have discussed the reviews with one another and the Reviewing Editor has drafted this decision to help you prepare a revised submission.

The reviewers found the idea of a checkpoint linking vacuole inheritance and/or biogenesis to the cell cycle potentially important. However, substantial discussion amongst the reviewers debated whether you have identified a bona fide cell growth checkpoint or simply confirmed that TORC1 signals from the vacuole membrane. It was appreciated that this may not be definitively distinguished at this stage and the reviewers have collectively offered a few experiments that they feel can raise the paper to the threshold of acceptance.

Essential revisions:

1) It wasn't clear from the data presented for the double mutants whether “G1 arrest” only refers to the DNA content, and whether these are unbudded cells or budded cells with unduplicated nuclei. This is an important distinction that needs to be resolved as it helps to form whether the signal is controlling multiple aspects of cell cycle coordination, or just one (i.e. chromosome duplication and/or cell growth and cytokinesis).

Similarly, the authors state in the Results and Discussion section that: “These observations strongly suggest that cells without a functional vacuole first arrest at G1 phase, and then eventually die.” This and the aforementioned issue could be addressed by making movies with the double mutant (rather than simply taking static pictures) with the Vph1-GFP and a second tagged marker (e.g. nuclear). From the data presented this should be easy for the authors to do.

2) When is Sch9 recruited to a newly formed vacuole? Figure 5 shows that there is a defect or delay in Sch9 recruitment to the vacuole of *vac17Δ* cells, but does not reveal when during the process of vacuole biogenesis this problem is resolved or how this relates to eventual bud growth. Does Sch9-2D3E localize to the vacuole, or does it bypass TORC1 signaling altogether?

Related, you may wish to address the following questions regarding TORC1 and Sch9 signaling that were raised during review:

The Weisman group has previously shown that PI(3,5)P_2_ on the vacuole membrane is required vacuole membrane targeting of Sch9 (35). The difference between the ‘new’ and maternal vacuoles in sustaining TORC1 signaling (and, therefore, cell growth) may reflect the acquisition of PI(3,5)P_2_ on the new vacuole membrane. Does the ‘new’ vacuole in *vac17* cells (Figure 4—figure supplement 1) contain PI(3,5)P_2_? It might also be informative to determine if synthetic targeting of Sch3-2D3E to the ‘old’ vacuole (e.g., by fusion to Vph1) rescue the cell growth defect?

3) The Barbet et al., MBoC, 1996 study (PMID: 8741837) on the role of Tor signaling in the regulation of cell cycle progression out of G1 should be cited and discussed.

---

## [Author Response]

*1) It wasn't clear from the data presented for the double mutants whether “G1 arrest” only refers to the DNA content, and whether these are unbudded cells or budded cells with unduplicated nuclei. This is an important distinction that needs to be resolved as it helps to form whether the signal is controlling multiple aspects of cell cycle coordination, or just one (i.e. chromosome duplication and/or cell growth and cytokinesis)*.

We tested and found that the *vac17∆ pep12-60*^*tsf*^ mutant arrests with 1N DNA content as unbudded cells. This data is now shown in new Figure 3 and presented in the text as follows: “Further evidence that this is a bona fide G1 arrest, came from the finding that the *vac17∆ pep12-60*^*tsf*^ mutant with 1N DNA content, arrests as unbudded cells. […] Together these results indicate that the vacuole is required for early G1 progression.” Note that at least 400 cells were counted for each experiment, and three independent experiments were performed (see Figure 3 legend).

*Similarly, the authors state: in the Results and Discussion section that: “These observations strongly suggest that cells without a functional vacuole first arrest at G1 phase, and then eventually die. This and the aforementioned issue could be addressed by making movies with the double mutant (rather than simply taking static pictures) with the Vph1-GFP and a second tagged marker (e.g. nuclear). From the data presented this should be easy for the authors to do*.

To gain additional insight into whether cells without a functional vacuole first arrest at G1 phase and then eventually die, we performed statistical analyses of the degree of G1 arrest compared with the degree of cell death. These new analyses are presented in the text as follows:

“Importantly, at 8, 12 and 24 hours after the shift to 37°C, the differences between the the *vac17∆ pep12-60*^*tsf*^ double mutant exhibited a statistically significant increase in G1 arrested cells (all p-values < 1x10^-3^). […] In wild-type cells, Whi5 nuclear localization is transient and released by Cdc28-Cln3 activity, which enables progression to early G1 phase.”

Based on these new analyses it did not appear essential to the current study to further analyze the relative timing of G1 arrest and cell death by performing movies of individual cells. In addition because the increase in cell death does not occur until at least 24 hr at 37°C, it would be difficult if not impossible to generate time-lapse movies of this length of individual fluorescently labeled cells of the v*ac17∆ pep12-60*^*tsf*^ double mutant, to determine whether all cells eventually die and to determine a time-course of cell death.

*2) When is Sch9 recruited to a newly formed vacuole?*
Figure 5
*shows that there is a defect or delay in Sch9 recruitment to the vacuole of* vac17Δ *cells, but does not reveal when during the process of vacuole biogenesis this problem is resolved or how this relates to eventual bud growth*.

We agree that this is a critical question. We showed there is less Sch9 on the newly formed vacuole in Figure 5. To directly address whether Sch9 is eventually recruited to the newly formed vacuole and when this occurs, we correlated the presence of fluorescent signals, for Vph1-CFP, Tor1-3xGFP and GFP-Sch9 (Figure 5). This new analysis shows that Sch9 recruitment to the newly formed vacuole is slower than the recruitment of Vph1 and Tor1. Specifically, small budded cells do not have fluorescent signals for any of the proteins indicating these small buds do not have a vacuole. In most cases, as the bud increases in a size, Vph1-CFP and Tor1-GFP appear simultaneously. This indicates that Tor1-GFP is immediately recruited to the newly formed vacuoles. In contrast, in the *vac17∆* mutant which co-expresses Vph1-CFP and GFP-Sch9, in medium sized buds (0.62(±0.14) daughter size/mother size) only Vph1-CFP is present. The average bud size where both Vph1 and Sch9 are present is 0.79(±0.11). Thus Sch9 recruitment is delayed compared to Tor1, but eventually occurs.

Does Sch9-2D3E localize to the vacuole, or does it bypass TORC1 signaling altogether?

The following text is now included in the manuscript: “It was previously shown that Sch9-2D3E localizes on the vacuole membrane ([59] Mol. Cell). […] This suggests that a functional vacuole is required for the roles of the TORC1-*SCH9* pathway in cell-cycle progression from G1 phase.”

*Related, you may wish to address the following questions regarding TORC1 and Sch9 signaling that were raised during review*:

*The Weisman group has previously shown that PI(3,5)P*_*2*_
*on the vacuole membrane is required vacuole membrane targeting of Sch9 (*[35]*). The difference between the ‘new’ and maternal vacuoles in sustaining TORC1 signaling (and, therefore, cell growth) may reflect the acquisition of PI(3,5)P*_*2*_
*on the new vacuole membrane. Does the ‘new’ vacuole in vac17 cells (*Figure 4—figure supplement 1*) contain PI(3,5)P*_*2*_*? It might also be informative to determine if synthetic targeting of Sch3-2D3E to the ‘old’ vacuole (e.g., by fusion to Vph1) rescue the cell growth defect?*

We thank the reviewers for this suggestion. To test whether the delay in Sch9 recruitment might be due to a delay in the generation of PI(3,5)P_2_ on the newly formed vacuoles, we tested the timing of the recruitment of Fab1, the sole lipid kinase for generation of PI(3,5)P_2_. Notably the timing of the recruitment of Fab1 was similar to that observed for Sch9. In a *vac17∆* mutant that co-expressed Vph1-CFP and Fab1-3xGFP, in medium sized buds (0.57(±0.13) daughter size/mother size) only Vph1-CFP is present. The average bud size where both Vph1 and Fab1 are present is 0.76(±0.10). Thus Fab1 recruitment is delayed compared to Tor1, and is similar to the recruitment of Sch9.

*3) The Barbet et al., MBoC, 1996 study (PMID: 8741837) on the role of Tor signaling in the regulation of cell cycle progression out of G1 should be cited and discussed*.

Thank you for this suggestion. We now cite and discuss [6] and state as follows:

“Together, these observations demonstrate that a functional vacuole is crucial for cell-cycle progression at G1 phase, and that the TORC1-*SCH9* pathway is part of this critical function (Figure 4). TORC1-*SCH9* signaling from the vacuole may be involved in G1 progression through its known functions in a ribosome biogenesis and translation (6; 38; 59)”.